# Unravelling Regioselectivity of *Leuconostoc citreum* ABK-1 Alternansucrase by Acceptor Site Engineering

**DOI:** 10.3390/ijms22063229

**Published:** 2021-03-22

**Authors:** Karan Wangpaiboon, Thassanai Sitthiyotha, Surasak Chunsrivirot, Thanapon Charoenwongpaiboon, Rath Pichyangkura

**Affiliations:** 1Department of Biochemistry, Faculty of Science, Chulalongkorn University, Bangkok 10330, Thailand; wangpaiboon9@gmail.com (K.W.); surasak.ch@chula.ac.th (S.C.); 2Structural and Computational Biology Research Unit, Department of Biochemistry, Faculty of Science, Chulalongkorn University, Pathumwan, Bangkok 10330, Thailand; mr_bri-an@hotmail.com; 3Department of Chemistry, Faculty of Science, Silpakorn University, Nakhon Pathom 73000, Thailand; charoenwongpaib_t@su.ac.th

**Keywords:** *Leuconostoc citreum*, alternansucrase, acceptor reaction, molecular dynamics simulation

## Abstract

Alternansucrase (ALT, EC 2.4.1.140) is a glucansucrase that can generate α-(1,3/1,6)-linked glucan from sucrose. Previously, the crystal structure of the first alternansucrase from *Leuconostoc citreum* NRRL B-1355 was successfully elucidated; it showed that alternansucrase might have two acceptor subsites (W675 and W543) responsible for the formation of alternating linked glucan. This work aimed to investigate the primary acceptor subsite (W675) by saturated mutagenesis using *Leuconostoc citreum* ABK-1 alternansucrase (*Lc*ALT). The substitution of other residues led to loss of overall activity, and formation of an alternan polymer with a nanoglucan was maintained when W675 was replaced with other aromatic residues. Conversely, substitution by nonaromatic residues led to the synthesis of oligosaccharides. Mutations at W675 could potentially cause *Lc*ALT to lose control of the acceptor molecule binding via maltose–acceptor reaction—as demonstrated by results from molecular dynamics simulations of the W675A variant. The formation of α-(1,2), α-(1,3), α-(1,4), and α-(1,6) linkages were detected from products of the W675A mutant. In contrast, the wild-type enzyme strictly synthesized α-(1,6) linkage on the maltose acceptor. This study examined the importance of W675 for transglycosylation, processivity, and regioselectivity of glucansucrases. Engineering glucansucrase active sites is one of the essential approaches to green tools for carbohydrate modification.

## 1. Introduction

Carbohydrates are an essential biomolecule which play important roles in biological functions (e.g., cell structure, cell recognition, and as energy sources). They are also compositional parts of other biomolecules. The study of carbohydrate structures is very difficult due to the high diversity of monosaccharides, the substitution of OH groups, and myriad linkage types. Unlike DNA, RNA and proteins, biosynthesis of carbohydrates is carried out without a template; this complicates the synthesis process [1]. Although automated oligosaccharide synthesis has been successfully developed (based on chemical synthesis), both the yield and the accuracy of regiochemistry are limited, including long time-consumption of the synthesis. [2,3]. To overcome these issues, enzymatic synthesis of carbohydrates could be developed, in order to take advantage of the high specificity and turnover rate. Moreover, the regiospecificity of carbohydrate-active enzymes is precisely controllable, as the enzymes have been continuously engineered via natural evolution. One carbohydrate type, α-glucan, has seen use in a broad range of food and health applications [4]. α-glucan can be synthesized by glucansucrases, most of which have belonged to glycoside hydrolase family 70 (GH70) [5]. Oligosaccharides and glucans synthesized from glucansucrases are economical in cost; their sole substrate is sucrose, or table sugar. Moreover, they can be scaled up via bacterial culturing in a specific medium or enzymatic synthesis. These factors make α-glucan very attractive for industrial applications—unlike several glycosyltransferases that require high-cost starting materials such as nucleotide sugar or phosphate sugar to facilitate product synthesis [6,7,8]. Recently, glucansucrases from a large number of bacterial sources have been explored, predominantly from lactic acid bacteria. Generally, glucansucrases can be classified into four main types based on their major linkage products: dextransucrase, mutansucrase, reuteransucrase, and alternansucrase, which produce α-1,6, α -1,3, α-1,4/1,6, and α-1,3/1,6 glucans, respectively [9]. Each α-glucan has unique properties which are dependent on linkage types. The sequence of linkage and molecular weight can also affect glucan properties [10,11,12]. 

Alternansucrase is a unique glucansucrases synthesized mainly by alternating α-1,3 and 1,6-linked glucans— thus, the use of the name alternan [13]. This glucan is water-insoluble, and spontaneously appears in solution as an opal nanocolloid with low viscosity in low concentration. On the other hand, a gel-like material can be obtained from a high concentration of alternan [14]. In addition, several novel oligosaccharides can efficiently be synthesized from alternansucrase via acceptor reaction. An acceptor reaction occurs when various sugars—e.g., isomaltose, maltose, nigerose, methyl α-d-glucoside [15], raffinose [16], gentiobiose, etc.,—are used as acceptor molecules [17]. In addition to the unusual physical properties of alternan, alternan that has been modified by sonication also exhibits biological activities which could enhance proliferation and migration of human mesenchymal stem cells via toll-like receptors [18]. Short-chain oligosaccharides produced by maltose-acceptor reactions in alternansucrase show potential as prebiotic agents [19].

The general structure of GH70 consists of five domains (A, B, C, IV, and V) of various glucansucrases. Alternansucrase also consists of seven SH3-like or APY domains at the C-terminal domain. These are found only in a select few glucansucrases [20]. The domains did not exhibit clearly specific roles in glucansucrase. However, deletion of partial and full SH3-like domains had a significant impact on the size of glucan nanoparticles and on the critical point of gel transformation. Overall characteristics of enzymes and products were slightly affected [21]. Recently, the crystal structure of truncated alternansucrase from *Leuconostoc citreum* NRRL B-1355 (ASR) has successfully been explicated. This structure could provide essential information on alternating linkage formation [22]. It became clear that the active site of alternansucrase was more complex than others (e.g., glucansucrases 180 (GTF180) [23] and dextransucrase (DSR-M)) [24]. Additionally, the ASR crystal structure and site-directed mutagenesis suggested that alternansucrase might contain two acceptor binding sites capable of participating in the synthesis of the alternating linked glucan. Like other carbohydrate-active enzymes, aromatic residues in glucansucrases were used for substrate recognition and binding [25,26]. W675 and W543 of alternansucrase were essential +2 and +3 subsites, respectively, suggesting that they provided an important hydrophobic stacking with acceptor sugar molecules. The W675 performed primary interaction with the acceptor molecule, whereas W345 conducted the next interaction, which was involved in the formation of the alternating linkage mechanism [22]. 

Although site-directed mutagenesis at the W675 position was reported, insight into this residue has not been completely investigated. In this study, we aimed to elucidate the effects of mutation at the W675 position by saturated mutagenesis using truncated *Leuconostoc citreum* ABK-1 alternansucrase (*Lc*ALT) [21]. We analyzed activity and kinetic parameters, including product characterization of all mutants. Finally, molecular dynamics (MD) simulations were employed to gain insight into how W675A mutations of *Leuconostoc citreum* ABK-1 alternansucrase caused the enzyme to produce products with various glycosidic linkages, while the wild-type ALT (WT) effectively produced oligoglucans with α-(1, 6) linkage via maltose-acceptor reaction.

## 2. Results

### 2.1. Effect of Substitution at W675 Position to Activity and Kinetic Study

Tryptophan residue was strictly conserved at the +2 binding subsite among various glucansucrases as shown in Figure 1. In this study, the +2 binding subsite of *Lc*ALT corresponded to W675 residue was substituted by F, Y, A, I, L, H, S, D, and N. All mutants were successfully expressed and purified (Appendix A). When the W residue at 675 of ALT was replaced, more than 50% of the specific activity was clearly lost. When F and Y (aromatic amino acids similar to W) were substituted, their specific activities were reduced by around 80 and 60% respectively, whereas those variants replaced by other amino acids lost at least 95% of wild-type activity, as shown in Figure 2. These results were correlated with the kinetic study. The *k_cat_* of the total, transglycosylation, and hydrolytic activity of all mutants apparently decreased compared to WT (Table 1).

### 2.2. Product Analysis

#### 2.2.1. Sucrose Reaction

The mutants substituted by aromatic residues (Y and F) still showed polymer products, while the other mutants produced oligosaccharides as a majority without polymer. This was shown in TLC and HPSEC results (Figure 3A and Figure 4). The low molecular weight (LMW) polymer peaks of W675Y and W675F at 5.75–7.5 min of HPSEC were significantly shifted from that of WT. Generally, the high molecular weight of the WT product can naturally form a nanoparticle, as described in previous work [14,21]. The hydrodynamic diameter (measured by DLS) of W675Y and W675F nanoparticles were 79.71 ± 0.33 and 82.07 ± 0.23 nm, while that of WT was 83.63 ± 0.83 nm (Appendix A). Additionally, ^1^H NMR spectra of WT, W675Y, and W675F polymers similarly exhibited a proportion of α-1,3 and α-1,6 linkages of approximately 40%:60% (Appendix A). The oligosaccharide patterns of WT and mutant enzymes were also analyzed by TLC and HPAEC-PAD. The oligosaccharide patterns of mutants substituted by A, I, L, H, S, D, and N were apparently different from WT. These mutant enzymes also produced short-chain oligosaccharides without polymer. On the other hand, the patterns of W675Y, and W675F were almost similar to that of WT (Figure 3A, Figure 4 and Appendix A).

#### 2.2.2. Maltose-Acceptor Reaction

When the maltose-acceptor reaction was conducted with ALTs, the major products were oligosaccharides. Small amounts of polymer products were found in WT, W675Y, and W675F (Figure 3B). The oligosaccharide pattern of W675Y and W675F corresponded with that of WT, while the other mutants clearly showed a distinguished pattern of oligosaccharides (Figure 3B and Appendix A). 

#### 2.2.3. Products Comparison between WT and W675A

Products from W675A were chosen to compare to those of WT. Di-to-pentasaccharides produced from WT sucrose reactions were composed of several species in individual mass, where di-to-pentamer isomalto-oligosaccharides were detected (Appendix A). The isomaltose and isomaltotriose were also found in W675A. Interestingly, nigerose could be found in the W675A disaccharide mixture while those of WT could not be observed. Moreover, W675A produced only a single species of tetrasaccharide from sucrose reaction as shown in Appendix A. 

For the maltose-acceptor reaction, tri-to-pentasaccharides from WT maltose-acceptor reactions (Appendix A) clearly exhibited a major peak in each size. Tri-to-pentasaccharides of W675A (Appendix A) represented several major peaks (Appendix A). The main peaks of WT trisaccharides corresponded with panose. Those of W675A contained a large number of peaks, of which several species corresponded with trisaccharides from W675A sucrose reaction (Figure 5). The partial hydrolysis of W675A trisaccharides released glucose, fructose, kojibiose, nigerose, maltose, and isomaltose. Those of WT generated glucose, maltose, and isomaltose, which were panose components, as shown in Figure 6. In contrast, the main tetrasaccharides from W675A showed an identical peak with that of WT (Appendix A).

### 2.3. Molecular Modelling and Simulations

Molecular dynamics were performed to gain insight into how W675A mutation of *Leuconostoc citreum* ABK-1 alternansucrase caused the enzyme to produce products with various glycosidic linkages, whereas WT effectively produces oligoglucans with α-(1, 6) linkage via maltose-acceptor reaction. The RMSD values of maltose-ALT_wt_ and maltose-ALT_W675A_ complexes were calculated to determine the stabilities of these systems during the simulations (Appendix A). All trajectories of these systems were used to measure the distances between O2, O3, O4 and O6 of the hydroxyl groups of the nonreducing end of maltose and the C1 atom of the glc-D635 intermediate (Figure 7). In terms of the maltose-ALT_wt_ complex, the distance between O6 of the nonreducing end and C1 of the glc-D635 intermediate was reasonable and quite stable (3.58 ± 0.34 Å). The distances between O2, O3 and O4 of the nonreducing end were too far from C1 of the glc-D635 intermediate for transglycosylation to synthesize α-(1, 2), α-(1, 3) and α-(1, 4) linkage of products. For maltose-ALT_W675A_ complex, the O2-C1, O3-C1, O4-C1 and O6-C1 distances were reasonable (≤3.5 Å) at points during the simulations. 

## 3. Discussion

Recently, the crystal structure of *Leuconostoc citreum* NRRL-B1355 alternansucrase (ASR))was successfully elucidated. The results suggested that the alternansucrase, producing mainly alternating α-(1, 3) and α-(1, 6)-linked glucans, might have two acceptor sites. W675 and W543 provided hydrophobic staking of each site [22]. This could reasonably explain its mechanism. The W675 was the main acceptor site, while the W543 was incorporated into another one. This work was interested in observing the impact of editing the W675 residue (the first acceptor site) in *Leuconostoc citreum* ABK-1 alternansucrase (*Lc*ALT). This position was highly conserved in many glucansucrases (Figure 1). Nonetheless, in some glucansucrases, this position was substituted by other residues. *Leuconostoc citreum* NRRL B-742 glucansucrase (BRS-B) produced a high ratio of α-(1, 3) branching dextran, since the +2 subsite was naturally substituted by P residue [27]. Meanwhile, characterization of the second catalytic domain of *Leuconostoc citreum* NRRL B-1299 glucansucrase (DSRE-CD2) showed that the enzyme exhibited low transglycosylation activity, but still maintained a high rate of hydrolytic activity—as its acceptor site was G residue. However, DSRE-CD2 might enhance to add α-(1, 2) branching to the glucan produced from the main catalytic domain (DSRE-CD1) [28,29].

Saturated mutagenesis was conducted at W675. The specific activity of the mutants substituted by other aromatic residues (F and Y) was apparently lost (approximately 60–80% compared with that of WT) (Figure 2). However, overall product patterns and polymerization activity remained similar to that of WT, though their linkage glucans, including nanoparticle product sizes, were slightly impacted. Although the *Lactobacillus reuteri* 180 glucansucrase (GTF180) produced glucan which showed a similar linkage pattern to that of alternan [14,30], the structure of its active site differed significantly from alternansucrase [22]. When the +2 acceptor subsite of GTF180 at W1065 was replaced by F residue, the glucan synthesis was clearly suppressed (approximately sevenfold); however, the W675F mutant of *Lc*ALT was hardly affected (Figure 3A and Figure 4). Furthermore, W1065F of GTF180 generated a small proportion of α-(1, 4) linkage in glucan [31], while glucan synthesized from W675Y and W675F of *Lc*ALT strictly exhibited linkages similar to those of the WT enzyme (Appendix A). 

Meanwhile, the specific activity of other variants, which were replaced by nonaromatic residues, were decreased by over 90%. This result correlated to another report in ASR where the acceptor site was substituted by A residue [22]. Moreover, polymer synthesis from the mutants was suppressed; the polymer could hardly be detected by TLC and HPSEC, indicating that the acceptor binding site was a crucial factor controlling processivity of the enzyme. All glucan binding domains remained in this truncated *Lc*ALT. Additionally, distinguishing oligosaccharide patterns obtained by these mutants were detected by TLC and HPAEC-PAD analysis (Figure 3A, Appendix A). These results corresponded with various W1065 mutants of GTF180 [31]. 

To confirm that the W675 position was an acceptor site of alternansucrase, the acceptor reaction was conducted. Several reports on various glucansucrases have demonstrated that maltose was an effective acceptor of glucansucrases [32,33,34,35]. The W675A variant was chosen for this demonstration. As a result, the maltose-acceptor reaction of W675A had contaminated products from sucrose reactions—according to fructose detection by partial hydrolysis (Figure 6)—that differed from the WT reaction. This did not occur in those of W1065 variants of GTF180 [31]. 

To gain insight into how W675A mutation of *Leuconostoc citreum* ABK-1 alternansucrase caused the enzyme to produce products with various glycosidic linkages (while WT effectively produced oligoglucans with α-(1, 6) linkage via maltose-acceptor reaction) molecular dynamics simulations were performed on the maltose-ALT_W675A_ and maltose-ALT_wt_ complexes. The results showed that WT could effectively control the orientation of maltose bound in the active site; therefore, O6 of the nonreducing end of maltose was close enough and in an appropriate orientation to attack C1 of the glc-D635 intermediate and form a panose molecule with α-(1, 6) linkage (Figure 7A). These findings supported results from experiments in WT’s maltose-acceptor reaction, which showed that panose was a major trisaccharide product (Figure 5 and Figure 6). W675 not only provided hydrophobic stacking to maltose (similar to those of GTF180 [23] and GTF-SI, [36]), but it also formed hydrogen bonds with maltose in WT. It did not form such bonds in the W675A mutant. These results indicated the importance of W675 in controlling the orientation of maltose binding in the active site of alternansucrase.

Moreover, molecular dynamics simulations showed that the W675A mutant could not tightly bind maltose in its active site. This site may have been competitively bound by other acceptors (such as sucrose) since fructose could be detected in the partial hydrolysis experiment (Figure 6). Consequently, the mutant enzyme could no longer effectively control the orientation of maltose. Therefore, maltose was more flexible and could easily change its orientation—such that its O2, O3, O4 and O6 had opportunities to come close to C1 of the glc-D635 intermediate and attack this atom to form products with various types of linkages (Figure 7B). These findings support experimental results of the partial hydrolysis of W675A trisaccharide products that detected kojibiose, nigerose, maltose, and isomaltose, which are disaccharides of homoglucose linked by α-(1, 2), α-(1, 3), α-(1, 4), and α-(1, 6). The reduced binding affinity to the substrate and decreased hydrophobicity at residue 675 of the W675A mutant may increase the chances of water attacking the acceptor site. The hydrolytic rate of the W675 mutant was dramatically increased, as shown by kinetic parameters in Table 1. These results were also observed in the +2 subsite mutation of GTF180 [31] and ASR [22]. Our findings revealed that this site was essential to transglycosylation, processivity, and regioselectivity of glucansucrases.

**Table 1 ijms-22-03229-t001:** Kinetic parameters of WT alternansucrase and its mutants substituted at W675 position.

	**Parameters**	**WT**			**W675Y**			**W675F**			**W675A**			**W675L**		
Total activity	K_m_ or k (mM)	25.5	±	6.1	14.2	±	2.0	16.6	±	2.1	43.0	±	4.7	46.4	±	6.0
	k_cat_ (s^−1^)	788.2	±	86.6	110.9	±	2.6	58.4	±	1.4	12.3	±	0.4	11.9	±	0.4
	k_cat_/K_m_ (s^−1^ mM^−1^)	30.9	±	14.1	7.8	±	1.3	3.5	±	0.6	0.3	±	0.1	0.3	±	0.1
Tranglycosylation	K_m_ or k (mM)	33.4	±	8.7	14.7	±	2.4	18.0	±	2.8	106.0	±	17.6	92.7	±	14.9
	k_cat_ (s^−1^)	815.6	±	112	91.1	±	2.6	47.1	±	1.4	9.9	±	0.6	9.2	±	0.6
	k_cat_/K_m_ (s^−1^ mM^−1^)	24.4	±	12.8	6.2	±	1.1	2.6	±	0.5	0.09	±	0.04	0.1	±	0.04
Hydrolysis	K_m_ or k (mM)	8.4	±	1.8	11.2	±	2.9	4.0	±	1.8	10.2	±	1.8	10.8	±	1.6
	k_cat_ (s^−1^)	56.0	±	1.6	19.6	±	0.8	11.2	±	0.4	3.8	±	0.1	3.5	±	0.1
	k_cat_/K_m_ (s^−1^ mM^−1^)	6.7	±	0.9	1.8	±	0.3	2.8	±	0.2	0.4	±	0.1	0.3	±	0.05
	**Parameter**	**W675I**			**W675H**			**W675S**			**W675D**			**W675N**		
Total activity	K_m_ or k (mM)	53.1	±	4.6	185.1	±	22.7	63.6	±	8.3	115.5	±	10.7	69.9	±	11.1
	k_cat_ (s^−1^)	7.3	±	0.2	1.6	±	0.2	10.3	±	0.4	2.4	±	0.1	6.8	±	0.4
	k_cat_/K_m_ (s^−1^ mM^−1^)	0.1	±	0.04	0.01	±	0.01	0.2	±	0.1	0.02	±	0.01	0.1	±	0.03
	Hill factor (n)				1.9		0.2									
Tranglycosylation	K_m_ or k (mM)	79.0	±	10.4	185.1	±	22.7	160.6	±	26.2	226.9	±	93.4	90.9	±	22.5
	k_cat_ (s^−1^)	5.0	±	0.2	1.6	±	0.2	9.0	±	0.7	1.7	±	0.5	4.1	±	0.6
	k_cat_/K_m_ (s^−1^ mM^−1^)	0.1	±	0.0	0.01	±	0.01	0.1	±	0.03	0.01	±	0.005	0.05	±	0.03
	Hill factor (n)				1.9		0.2				1.4		0.3	1.5		0.4
Hydrolysis	K_m_ or k (mM)	20.3	±	4.9	185.1	±	22.7	10.6	±	1.9	6.9	±	2.2	2.9	±	1.9
	k_cat_ (s^−1^)	2.4	±	0.1	0.01	±	0.001	2.9	±	0.1	0.8	±	0.03	1.9	±	0.1
	k_cat_/K_m_ (s^−1^ mM^−1^)	0.1	±	0.02	0.00004	±	0.00003	0.3	±	0.04	0.1	±	0.01	0.7	±	0.04
	Hill factor (n)				1.9		0.2									

## 4. Materials and Methods 

### 4.1. Mutagenesis and Enzyme Preparation

Mutagenesis was performed by PCR-driven overlap extension method [37]. The PCR products were amplified by PrimSTAR^®®^ DNA polymerase (TAKARA, Shiga Japan) and used pETΔ7SH (wild-type, WT) plasmid [21] as a template. The primer list is shown in Appendix A. The PCR products were cloned into pET19b vectors via NcoI and XhoI restriction endonuclease site and transformed into E. coli TOP10. The transformants were randomly picked and analyzed by restriction enzyme analysis. Finally, the sequences of positive clones were confirmed by DNA sequencing (1stBase DNA sequencing, Malaysia). Expression and purification were provided as described in the previous work [21].

### 4.2. Enzymatic Assay

The enzyme was incubated with 20% (*w*/*v*) sucrose and 50 mM sodium citrate buffer pH 4.0 at 40 °C for 15 min in 0.5 mL total volume. The reaction was stopped by adding an equal volume of 3, 5-dinitrosalisylic acid (DNS) solution [38], and then the reaction was boiled for 10 min. The absorbance of the reaction was monitored at A_540_. Fructose was used as a standard for the calibration curve. One unit of the enzyme was defined as the amount of enzyme that can produce 1 μmol of fructose in one min.

### 4.3. Kinetic Study

The kinetic study was conducted in 0–368 mM sucrose and 50 mM sodium citrate buffer pH 4.0 at 40 °C in 0.5 mL total volume. The reactions were initiated by the enzymes, after which the reactions were terminated by adding 35 μL of 1 N NaOH. Each reaction was split for DNS and glucose oxidase assay (Glucose LiquiColor^®®^, Human, Wiesbaden, Germany). Total activity was defined by the total reducing sugar of DNS assay. Transglycosylation activity was measured as the difference between total reducing sugar and free glucose content, while hydrolytic activity corresponded to total free glucose. The kinetic data of all enzymes was fitted and analyzed using OriginPro2017 software. All kinetic experiments were conducted in three replications.

### 4.4. Product Production

#### 4.4.1. Sucrose Reaction

The reactions comprised 200 mM sucrose, 50 mM sodium citrate buffer pH 4.0 and 0.1 U/mL enzyme. The reactions were incubated at 37 °C for 20 h. 

#### 4.4.2. Maltose-Acceptor Reaction

The reactions were conducted in 100 mM maltose, 100 mM sucrose and 0.1 U/mL enzymes in 50 mM sodium citrate buffer pH 4.0. Next, the reactions were incubated at 37 °C for 20 h. 

#### 4.4.3. Polymer Preparation

The reaction-produced polymers were precipitated by an equal volume of acetone. The precipitates were completely resuspended in distilled water and reprecipitated 3 times before freeze-drying.

### 4.5. Product Characterizations

#### 4.5.1. Thin-Layer Chromatography (TLC) Analysis

The samples were spotted on a TLC plate (TLC Silica gel 60 F_254_, Merck, Darmstadt, Germany). The TLC was run for 3 ascents in the TLC tank equilibrated with 1-butanol:acetic acid:water (3:3:2) for at least 48 h. Finally, the product patterns were monitored after spraying with Orcinol solution and incubated at 110 °C for 10 min. 

#### 4.5.2. High-Performance Anion-Exchange Chromatography with Pulsed Amperometric Detection (HPAEC-PAD) Analysis

The samples were analyzed by HPAEC-PAD using CarboPac^TM^ PA1 column (4 × 250 mm). The column was eluted by a linear gradient of 500 mM sodium acetate in 150 mM sodium hydroxide for 30 min, and then the concentration was held for 5 min with a flow rate of 1 mL/min. Maltooligosaccharides (α-1,4-linked G1-G7, ( Sigma–Aldrich, Darmstadt, Germany) and (HAYASHIBARA, Okayama, Japan)), isomaltose (TCI, Saitama, Japan), isomaltotriose (TCI, Saitama, Japan), nigerose (Sigma, USA), kojibose (Sigma–Aldrich, Darmstadt, Germany), panose (HAYASHIBARA, Okayama, Japan), leucrose (Sigma–Aldrich, Darmstadt, Germany), glucose (Ajax Finechem, Taren Point, Australia), fructose (Sigma–Aldrich, Darmstadt, Germany), and sucrose (Ajax Finechem, Taren Point, Australia), were used as standards. On the other hand, α-1,6-linked glucooligosaccharides were prepared by partial hydrolysis of dextran M.W.500 kDa (GE-Healthcare, Uppsala, Sweden).

#### 4.5.3. High Pressure Size Exclusion Chromatography (HPSEC) Analysis

The product reactions were analyzed by an OHpak SB-805 HQ column (6 × 300 mm) at 40 °C with a flow rate of 1 mL/min. The peak signal was monitored by a refractive index (RI) detector.

#### 4.5.4. Separation of Oligosaccharides

Approximately 350 mg crude products from sucrose or maltose-acceptor reaction (described as above) were loaded onto a Bio-Gel P-2 (Bio-Rad) column (5 × 120 cm). The samples were eluted by distilled water with a flow rate of 0.6 mL/min at 50 °C. The fractions were collected every 3.5 mL. All fractions were spotted on TLC and then sprayed by Orcinol solution. The bands were visualized after heating at 110 °C for 10 min. The band intensity was measured by Quantity One software (Bio-Rad), and the information was plotted as an elution profile. The pattern and size were primarily analyzed by TLC and subsequently confirmed by HPAEC-PAD and matrix assisted laser desorption ionization coupled with time of flight mass spectrometer (MALDI-TOF MS). 

#### 4.5.5. Nuclear Magnetic Resonance Spectroscopy (NMR) Analysis

Pretreated glucans by ultrasonication (as described in previous work [14]) were dissolved in D_2_O and ^1^H spectrum was recorded by BRUKER AVANCE III HD/OXFORD 500 MHz.

#### 4.5.6. Dynamic Light Scattering (DLS) Analysis

One mg/mL purified glucans were dispersed in distilled water and subjected to filtrate by 0.22 μm filter. The hydrodynamic diameter of nanoglucans was determined by DLS (Malvern Nanosizer ZS) at 25 °C. The size was determined for three replicates.

#### 4.5.7. Product Analysis by Partial Hydrolysis

The mixture solution of trisaccharides was boiled in 1 M HCl for 30 min. After that, the reaction was neutralized by equivalent NaOH. The reaction was then desalted by adding AG^®®^ 501-X8 mixed bed resin (BIO-RAD) before further analysis by HPAEC-PAD.

### 4.6. Molecular Dynamics Simulations

The structure of maltose was taken from the crystal structure of *Lactobacillus reuteri* glucansucrase (PDB: 3KLL [23]). It was solvated in an isomeric truncated octahedral box of TIP3P water and minimized using AMBER18 [39]. The SWISS-MODEL server [40,41,42,43] was employed to build the homology model of *Leuconostoc citreum* ABK-1 alternansucrase based on the crystal structure of *Leuconostoc citreum* NRRL B-1355 Alternansucrase (PDB: 6HVG [22]), which has the highest sequence identity—97.40% to the target sequence. The H^++^ server [44] was used to protonate all ionized amino acids at the experimental pH of 4.0. To construct the structure of glucosyl-D635 (glc-D635) intermediate, the structure of glc-D635 intermediate was obtained from the crystal structure of *Neisseria polysaccharea* amylosucrase (PDB: 1S46 [45]). The Gaussian09 program [46] and the Antechamber module of AMBER18 were used to generate charges and parameters of glc-D635 intermediate. LEaP module was employed to construct the structure of alternansucrase with glc-D635 intermediate. It was minimized with the five-step procedure [47,48,49,50,51,52,53]. Then, the system was heated, equilibrated and subsequently simulated for the production run at the experimental temperature of 310 K for 70 ns. Root mean square deviation (RMSD) values were calculated to monitor the structural stability of the system. The last 20 ns trajectories with stable RMSD values were selected for structural clustering by MMTSB toolset [54] based on their structural similarities as analyzed by RMSD values of heavy atoms. The structure that was the most similar to the average structure was selected as a centroid (a representative structure) for further analyses.

To predict the catalytically competent binding conformations of maltose in the binding site of *Leuconostoc citreum* ABK-1 alternansucrase, the Vina-Carb program [55] was used to dock maltose into the active site of the centroid structure of alternansucrase with the glc-D635 intermediate. In total, 20 independent docking runs were performed. The catalytically competent binding conformations were selected if the distance between the O6 hydroxyl group of the nonreducing terminal glucose of maltose and the C1 atom of the glc-D635 intermediate was less than or equal to 3.5 Å. [22] The catalytically competent binding conformations that passed the distance criterion were then superimposed with the crystal structure of maltose in the acceptor binding site of *Lactobacillus reuteri* 180 glucansucrase (PDB: 3KLL). The binding conformation with the lowest heavy-atom RMSD values was determined to be the maltose-alternansucrase (maltose-ALT_wt_) complex. W675 of the maltose-ALT_wt_ complex was mutated to A675 to construct the maltose-ALT_W675A_ complex. The maltose-ALT_wt_ and maltose-ALT_W675A_ complexes were immersed in an isomeric truncated octahedral box of TIP3P water molecules with a buffer distance of 13 Å and neutralized by sodium ions (Na^+^). All systems were minimized via the five-step procedure and then heated, equilibrated and simulated for 100 ns of production runs with similar setup procedures.

RMSD values with respect to the minimized structure were calculated to monitor the structural stability during the simulations of all systems. The proximity between atoms necessary for transglycosylation was measured as the distances between O2, O3, O4 and O6 of the hydroxyl groups of the nonreducing terminal glucose of the maltose and the C1 atoms of the glc-D635 intermediate (O2C1, O3-C1, O4-C1 and O6-C1 distances). Hydrogen bond interactions were calculated by hydrogen bond occupations between binding residues and maltose. Hydrogen bonds were considered to have occured if the following criteria were met: (i) a proton donor-acceptor distance ≤3.5 Å and (ii) a donor-H-acceptor bond angle ≥120°.

## 5. Conclusions

Our study unraveled the role of the acceptor binding site of *L. citreum* ABK-1 alternansucrase. Our results demonstrated that W675 of alternansucrase was strictly conserved. This position could not be substituted by other residues—even other aromatic residues. Replacement by other residues at the W675 position led to defective interactions with the acceptor molecule. Although substitution of the alternansucrase acceptor site led to decreases in activity, the gain of new product patterns may have been worthwhile compensation. The decoration of glucansucrase active sites could be an attractive approach to the generation of new catalysts that could potentially be used in carbohydrate synthesis.

## Figures and Tables

**Figure 1 ijms-22-03229-f001:**
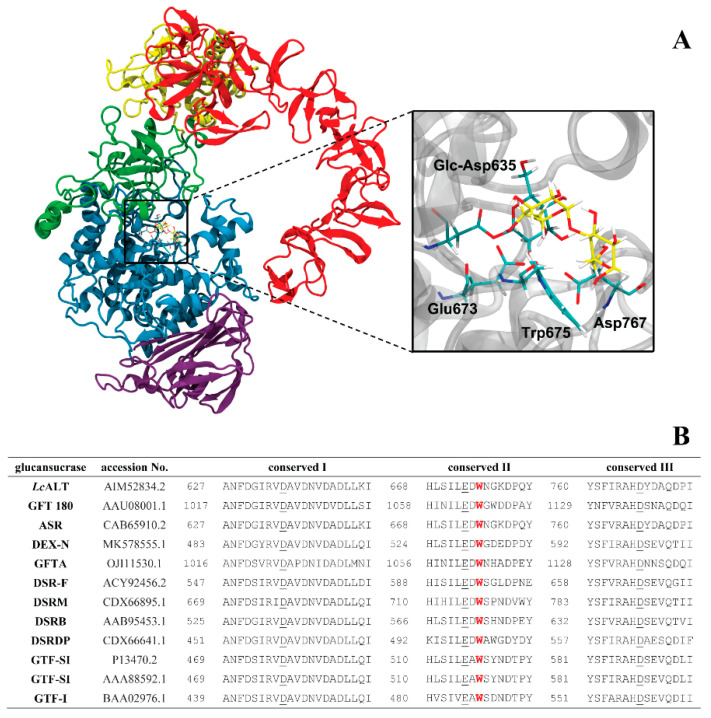
Homology model and conserved sequence of various glucansucrase. (**A**) The catalytically competent binding conformations of maltose in the binding site of *Leuconostoc citreum* ABK-1 alternansucrase, containing the glucosyl-D635 (glc-D635) intermediate. (**B**) Three conserved sequences of various glucansucrases. The underlined residues are catalytic residues, while the red residue is the conserved +2 tryptophan subsite in *Lc*ALT and other glucansucrases.

**Figure 2 ijms-22-03229-f002:**
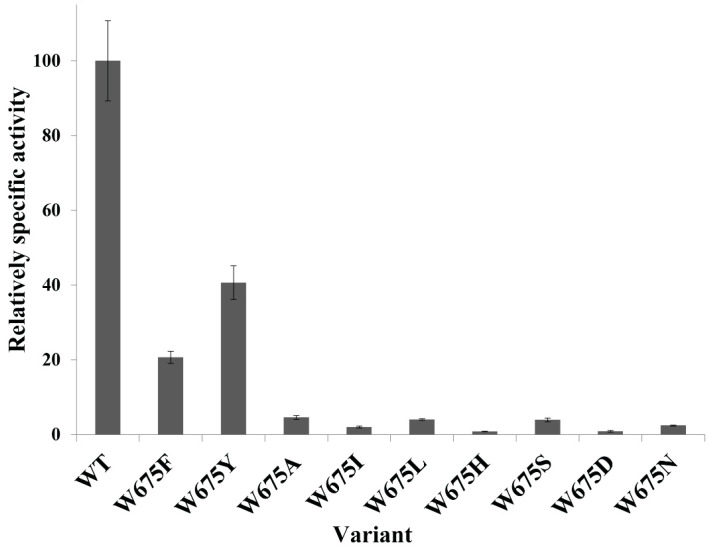
Comparison of relatively specific activity. All substituted mutants at W675 and WT alternansucrase were compared with specific activity. Activity assayed by DNS method. Protein content measured by Bradford assay.

**Figure 3 ijms-22-03229-f003:**
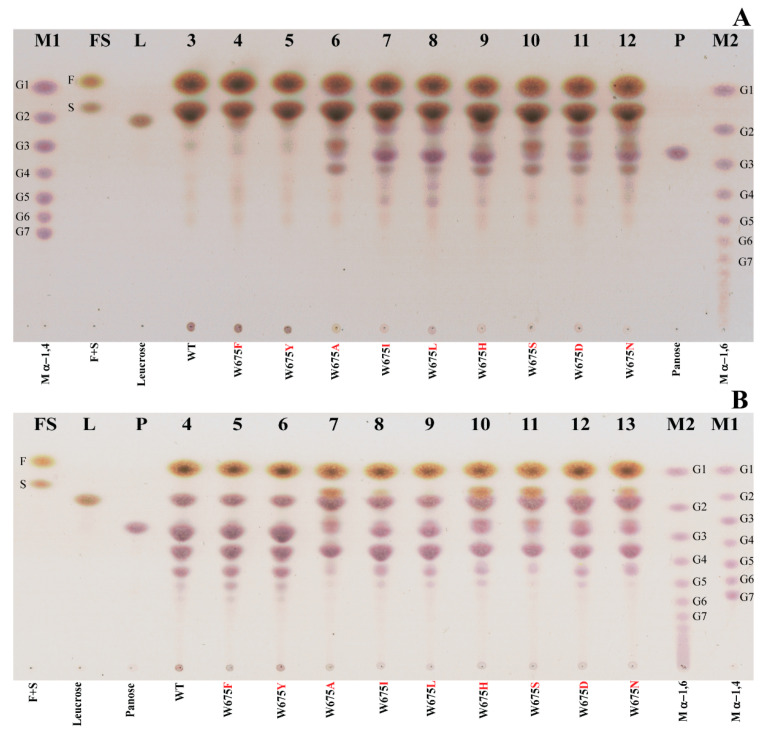
TLC analysis of WT and W675 mutants. (**A**) Products produced from sucrose reactions. (**B**) Products of maltose-acceptor reactions. Lanes M1 and M2—α-1,4 and α-1,6 glucooligosaccharides markers; Lane FS—std. fructose and sucrose; Lane 2—std. leucrose; Lane 13—panose; Lanes 3-12—products from WT, mutants substituted by F, Y, A, I, L, H, S, D, and N, respectively.

**Figure 4 ijms-22-03229-f004:**
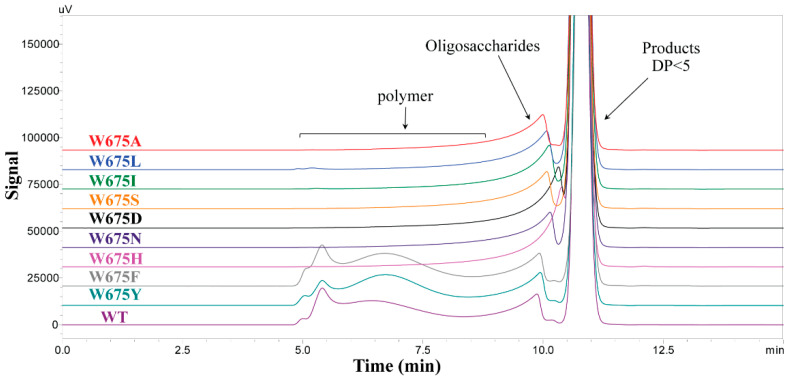
HPSEC analysis of sucrose reaction. The overnight reactions from mutants and WT alternansucrase were analysed using an OHpak SB-805 HQ column and monitored by RI detector.

**Figure 5 ijms-22-03229-f005:**
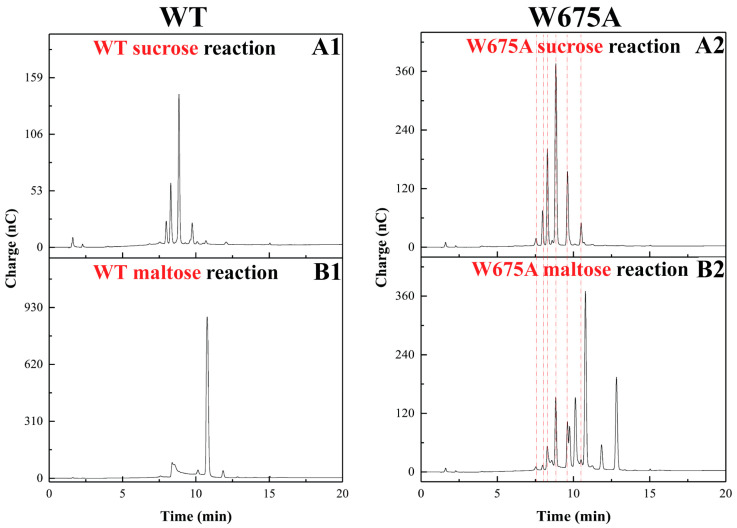
Comparison of trisaccharides from WT and W675A. Purified trisaccharides from sucrose and maltose acceptor reaction as analyzed by HPAEC-PAD. Mass verified by MALDI-TOF MS. (**Left panel**): trisaccharides from WT alternansucrase. (**Right panel**): W675A. (**A**) Trisaccharides from sucrose reaction. (**B**) Trisaccharides from maltose-acceptor reaction. Red dash lines indicate overlapping products from W675A sucrose and maltose reactions.

**Figure 6 ijms-22-03229-f006:**
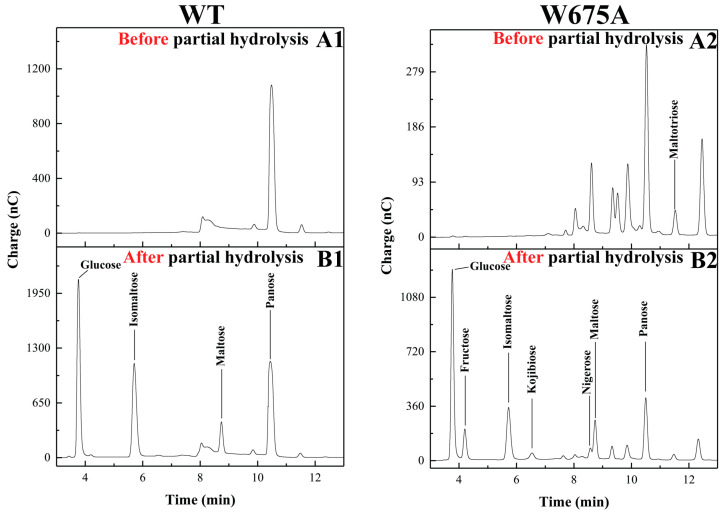
Partial hydrolysis of trisaccharides from maltose-acceptor reaction. (**Left panel**): products produced from WT. (**Right panel**): products from. (**A**) HPAEC-PAD profiles of the products before partial hydrolysis. (**B**) Product profiles after partial hydrolysis.

**Figure 7 ijms-22-03229-f007:**
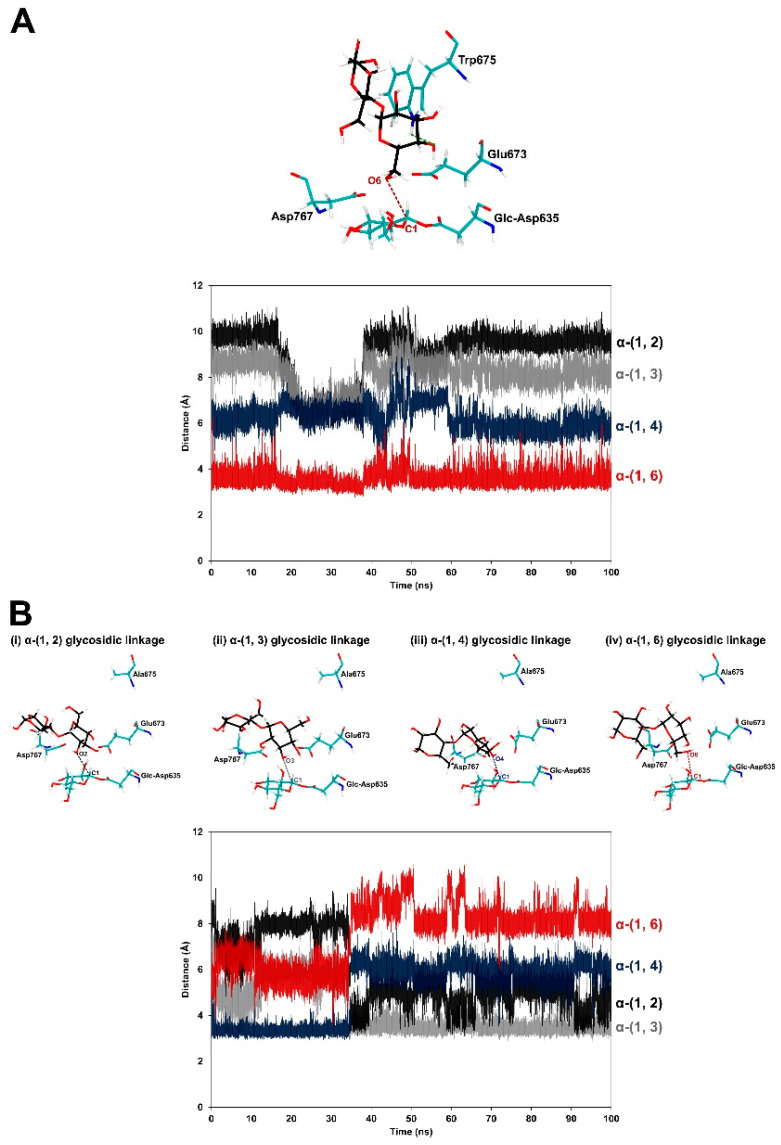
The proximity between atoms necessary for transglycosylation and hydrogen bond interaction of (**A**) maltose-ALT_wt_ and (**B**) maltose-ALT_W675A_ complexes. The distances between O2, O3, O4, O6 of the nonreducing terminal glucose of maltose and the C1 atom of the glc-D635 intermediate are shown in black, grey, blue, and red, respectively. Hydrogen bond interactions are represented with a green dashed line.

## Data Availability

Not applicable.

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
