# Peer review of "Unravelling Regioselectivity of Leuconostoc citreum ABK-1 Alternansucrase by Acceptor Site Engineering"

_ijms, 2021, doi:10.3390/ijms22063229_

Round 1

Reviewer 1 Report

The publication entitled: Unravelling regioselectivity of Leuconostoc citreum ABK-1 al-2 ternansucrase by acceptor site engineering looks quite desirable but requires a few corrections.

Below are my detailed comments.

  1. Why is Karan Wangpaiboon yellow? Did the author have any particular influence on the publication?
  2. You can gently change the tip of the abstract so that it more clearly indicates what the work has been done.
  3. I would treat the keywords in more detail - because the job seems to be important.
  4. "... cell recognition, and energy source" - you need to add something more. This is not their only role, is it?
  5. "-1,4 / 16, and -1,3 / 1,6" line 50, page 2 - is it good here?
  6. The introduction should be improved - there is a bit of a pity.
  7. We must add citations - there are definitely too few of them considering what appears in the literature.
  8. The end of the introduction needs to be corrected - what is it anyway? Authors must emphasize the purpose of their work more clearly.
  9. Section: 2.5.6. NMR analysis, 2.5.7. Dynamic light scattering (DLS) analysis, they need to be corrected - we don't know how it was done.
  10. Figure 2 - the signature of the x axis is to be improved.
  11. Figure 4 - the signature of the x axis is to be improved. Seems disproportionate.
  12. There is almost no summary. Please considerably improve the conclusion section. And separate her.

In conclusion, I think that the job is really good and my comments are only more corrective than substantive.

However, they affect the quality of the work, so the authors should refer to them. I recommend major revision and would like to see some bug fixes work.

Author Response

The publication entitled: Unravelling regioselectivity of Leuconostoc citreum ABK-1 alternansucrase by acceptor site engineering looks quite desirable but requires a few corrections.

Below are my detailed comments.

  1. Why is Karan Wangpaiboon yellow? Did the author have any particular influence on the publication?

Response: It was nothing special. The journal staffs made a new format for our manuscript. They just reminded and concerned us to check spelling for “Karan Wangpaiboon” name.

  1. You can gently change the tip of the abstract so that it more clearly indicates what the work has been done.

Response: We added “Engineering of glucansucrase active site is one of essential approaches to serve green tools for carbohydrate modifying.” in the end of abstract. (line24-25)

  1. I would treat the keywords in more detail - because the job seems to be important.

Response: We edited the keywords as Leuconostoc citreum; alternansucrase; acceptor reaction; molecular dynamics simulation.(line26)

  1. "... cell recognition, and energy source" - you need to add something more. This is not their only role, is it?

Response: We added “including being compositions of other biomolecules” in the sentence. (line30-31)

  1. "a-1,4 / 16, and a-1,3 / 1,6" line 50, page 2 - is it good here?

Response: In our opinions, these words are appropriate since these glucans have mixed linkages and inconsistently alternating linkage patterns are not consistent.

  1. The introduction should be improved - there is a bit of a pity.

Response: Our introduction was comprised with general carbohydrate, glucansucrase, alternansucrase, structure and mechanism, and purpose statements in paragraph 1-4, respectively. We don’t know exactly how the introduction should be improved. If the reviewer still thinks, it should be improved more. Could the reviewer specifically indicate which topics/paragraphs/lines should be improved ? However, we edited following question No.4. (line30-31) and also correct some gramma mistakes.

  1. We must add citations - there are definitely too few of them considering what appears in the literature.

Response: We have known about this point but there are a few research group in the world working with alternansucrase and have had only one alternansucrase structure deposited in data base.

  1. The end of the introduction needs to be corrected - what is it anyway? Authors must emphasize the purpose of their work more clearly.

Response: We already informed the statement of purpose “In this study, we aim to elucidate the effects of the mutation at W675 position by saturated mutagenesis using truncated Leuconostoc citreum ABK-1 alternansucrase (LcALT)”. (line87-89)

  1. Section: 2.5.6. NMR analysis, 2.5.7. Dynamic light scattering (DLS) analysis, they need to be corrected - we don't know how it was done.

Response: We could not understand well the reviewer’s question. The methods in 2.5.6 and 2.5.7 are common protocols and details for these techniques. If the reviewer still thinks that they miss some important information, please indicate our mistakes and let us know.

  1. Figure 2 - the signature of the x axis is to be improved.

Response: it was improved as suggested.

  1. Figure 4 - the signature of the x axis is to be improved. Seems disproportionate.

Response: We already added the axis labels.

  1. There is almost no summary. Please considerably improve the conclusion section. And separate her.

Response: We improved more as suggested. (line408-409)

 In conclusion, I think that the job is really good and my comments are only more corrective than substantive.

However, they affect the quality of the work, so the authors should refer to them. I recommend major revision and would like to see some bug fixes work.

Reviewer 2 Report

The manuscript is devoted to site-directed mutagenesis of alternansucrase from Leuconostoc citreum. A number of mutants were obtained, and products of enzymatic reaction were characterized. Unfortunately, the manuscript written is extremely confusing, that make it difficult to understanding. Nevertheless, the manuscript contains new interesting data on enzymes.

Authors have analyzed the composition of low-molecular products of the enzymatic reaction in detail, however, structure of high-molecular weight products unfortunately is poor established. According to Fig. 4 HMW are the main products for WT, W675Y and W675F. In my opinion, it is necessary to calculate the yield of low- and high-molecular weight products (HMW) and structure of HMW should be elucidated by NMR or/and mass spectrometry.

Authors can improve the manuscript in accordance with suggestions as followed:

Section “Materials and Methods”

What kind of citrate buffer was used? It was Na-citrate K-citrate or phosphate-citrate buffer, please, indicate.

line 106. In my opinion this paragraph can be divided into the two parts: 2.2. Hydrolytic activity assay and 2.3. Transglycosylating activity assay (Authors can move sentence from line 117 and add information about incubation mixture for transglycosylating activity investigation)

Section “Results”

line 248 “These mutant enzymes…” Authors should reference on figure 4

line 250 Please replace “(Fig.2 and S4) by (Fig. 3a, Fig. 4)”. Figure S4 is not informative because contains many not identified peaks. Authors must explain in footnotes of S4 figure what does it means the main peaks.

Figure 3. Authors should indicate degree of polymerisation of M1 and M2 standards, and include in “Materials and methods” section methods to obtaining or where it was purchase.

line 261 please replace “3.2.2 maltose-..” by “3.2.2 Maltose-…”

Figure S6. Please, explain numbers on TLC what’s they mean? Resolution of MALDI TOF spectra is not satisfactory, authors should improve it.

Figure 5. It contains more than one peak, which one belong to trisaccharide, please, indicate.

Section 3.2.3. and Figure 5. Why authors have compared products of the action WT and W675A? I was confused. Please, explain.

Author Response

Reviwer2

Comments and Suggestions for Authors

The manuscript is devoted to site-directed mutagenesis of alternansucrase from Leuconostoc citreum. A number of mutants were obtained, and products of enzymatic reaction were characterized. Unfortunately, the manuscript written is extremely confusing, that make it difficult to understanding. Nevertheless, the manuscript contains new interesting data on enzymes.

  1. Authors have analyzed the composition of low-molecular products of the enzymatic reaction in detail, however, structure of high-molecular weight products unfortunately is poor established. According to Fig. 4 HMW are the main products for WT, W675Y and W675F. In my opinion, it is necessary to calculate the yield of low- and high-molecular weight products (HMW) and structure of HMW should be elucidated by NMR or/and mass spectrometry.

Response: We tried to check the peak area of them. They are different just a few approximately 0.3-0.4 folds that of WT. Moreover, the calculation is not accurate for this case since the resolution of peaks are overlapped between HMW and LMW. The structure of alternan polymer was studied in previous work (wangpaiboon et al, 2018).

Authors can improve the manuscript in accordance with suggestions as followed:

Section “Materials and Methods”

  1. What kind of citrate buffer was used? It was Na-citrate K-citrate or phosphate-citrate buffer, please, indicate.

Response: We replaced it by “sodium citrate” at all.

  1. line 106. In my opinion this paragraph can be divided into the two parts: 2.2. Hydrolytic activity assay and 2.3. Transglycosylating activity assay (Authors can move sentence from line 117 and add information about incubation mixture for transglycosylating activity investigation)

Response: In the part of general assay (section 2.2), the activity was defined and determined by total activity (transglycosylation + hydrolytic activiy) as following “One unit of the enzyme was defined as the amount of enzyme can produce 1 mmol of fructose in one min.”

Section “Results”

  1. line 248 “These mutant enzymes…” Authors should reference on figure 4

Response: We added as suggested and also correct the order of all reference figure Number.

  1. line 250 Please replace “(Fig.2 and S4) by (Fig. 3a, Fig. 4)”. Figure S4 is not informative because contains many not identified peaks. Authors must explain in footnotes of S4 figure what does it means the main peaks.

Response: We edited as suggested. The main peaks; glucose, fructose, and sucrose, were already indicated.

  1. Figure 3. Authors should indicate degree of polymerisation of M1 and M2 standards, and include in “Materials and methods” section methods to obtaining or where it was purchase.

Response: We added the information as suggested. (line145-151)

  1. line 261 please replace “3.2.2 maltose-..” by “3.2.2 Maltose-…”

Response: We corrected it.

  1. Figure S6. Please, explain numbers on TLC what’s they mean? Resolution of MALDI TOF spectra is not satisfactory, authors should improve it.

Response: We apologized for our mistakes. We improved them by adding the labels and axis names. For the resolution of MALDI TOF, they could not be edited not much since they were a format exported from the instrument software. However, we labeled the desired masses.

  1. Figure 5. It contains more than one peak, which one belong to trisaccharide, please, indicate.

Response: The all peaks were trisaccharides since they were separated from size-exclusion column and then confirmed by MALDI-TOF MS. We also added the dash lines indicating the overlapping peaks between products from sucrose and maltose reactions.

  1. Section 3.2.3. and Figure 5. Why authors have compared products of the action WT and W675A? I was confused. Please, explain.

Response: The alanine residue is a commonly substituted residue for observing role of any residues by site-directed mutagenesis because the side chain of alanine is a methyl group lacking any influences such as charge, polar, and steric effects. For example, Charoenwongpaiboon et al., 2019.

Reviewer 3 Report

The study described in this manuscript covers an interesting topic and provides interesting results. There are just some minor points which should be considered.

Line 50: alpha-1,4/1,6

Line 105-111: Were Blank samples (only sucrose) analyzed? Boiling sucrose at pH 4 most likely results in some inversion to glucose and fructose.

Line 264: Fig. 5 is not the Figure to cite here, as it does not show polymers, but the trisaccharide fraction from BioGel fractionation.

Line 266: Only Fig. S5 should be cited here.

Fig. 5: Were there other masses which point to other DPs? There seem to be other mass peaks in the supplementary figures. Maybe add to the caption that BioGel fractionation was carried out and maybe cite Fig. S6 & Fig. S7 in the caption as they provide additional information related to this figure.

Fig. S6: Add axis labels and a legend for the TLC lanes.

Line 276: The synthesis of several solely 1,6-linked isomalto oligosaccharides is interesting because one would expect from an alternansucrase that it would only synthesize alternan oligosaccharides. This should be discussed.

Line 278: It should be mentioned that the relative abundance / the amount of nigerose is very low.

Line 282: Maybe cite Fig. S9 & Fig. S10 here, they provide useful information.

Fig. S10: This figure shows the results from W675A, right? Correct caption.

Fig. 6: Several mistakes in the caption.

At several points, the language should be improved (e.g. line 16, 30/31, 35, 75, 85, 103, 109, 284, 319, 336, 376 and several other occasions).

Author Response

The study described in this manuscript covers an interesting topic and provides interesting results. There are just some minor points which should be considered.

 Line 50: alpha-1,4/1,6

Response: We corrected it

  1. Line 105-111: Were Blank samples (only sucrose) analyzed? Boiling sucrose at pH 4 most likely results in some inversion to glucose and fructose.

Response: It was sure that blank was included in analysis. Actually, the enzyme activity was terminated by DNS solution prior boiling. The DNS is an alkaline solution (pH ~14). So, the hydrolysis by acid catalysis could not be occurred.

  1. Line 264: Fig. 5 is not the Figure to cite here, as it does not show polymers, but the trisaccharide fraction from BioGel fractionation.

Response: We corrected the Fig number and replaced into Fig 3B.

  1. Line 266: Only Fig. S5 should be cited here.

Response: We corrected it.

  1. 5: Were there other masses which point to other DPs? There seem to be other mass peaks in the supplementary figures. Maybe add to the caption that BioGel fractionation was carried out and maybe cite Fig. S6 & Fig. S7 in the caption as they provide additional information related to this figure.

Response: We added the fraction numbers on the MS Figures. The other masses were a background of the system and environment even our samples were not added, since the MALDI-TOF is very high sensitivity.

  1. S6: Add axis labels and a legend for the TLC lanes.

Response: We edited as suggested

  1. Line 276: The synthesis of several solely 1,6-linked isomalto oligosaccharides is interesting because one would expect from an alternansucrase that it would only synthesize alternan oligosaccharides. This should be discussed.

Response: In fact, the linkage pattern of alternan polymer did not show constantly alternating sequences as shown in previous our study (Wangpaiboon et al., 2018). It has isomalto-oligosaccharide pattern inserted inside the polymer.

  1. Line 278: It should be mentioned that the relative abundance / the amount of nigerose is very low.

Response: Because the oligosaccharides were analysed by HPAEC detected with PAD (Electrochemical detector). In general, the signals from different species detected by electrochemical detector could not be compared. So, it could not be compared among different species products.

  1. Line 282: Maybe cite Fig. S9 & Fig. S10 here, they provide useful information.

Response: We cited as suggested.

  1. S10: This figure shows the results from W675A, right? Correct caption.

Response: That was our mistake. It was corrected.

  1. 6: Several mistakes in the caption.

Response: We corrected.

  1. At several points, the language should be improved (e.g. line 16, 30/31, 35, 75, 85, 103, 109, 284, 319, 336, 376 and several other occasions).

Response: The incorrected sentences/words as mention lines were repaired. The other mistakes also corrected as shown in yellow.

Round 2

Reviewer 1 Report

The authors significantly improved the work and responded very nicely to my objections.
The work is the most fun and worth publishing in IJMS.
I recommend publishing this work.

Reviewer 2 Report

The authors took into account all my suggestions, I recommend  this manuscript  for publication.